# Development and preclinical evaluation of a cable-clamp fixation device for a disrupted pubic symphysis

Martin C. Jordan [1✉], David Bröer[1], Christian Fischer[2], Philipp Heilig [1], Fabian Gilbert[3],
Stefanie Hölscher-Doht[1], Charis Kalogirou[4], Kevin Popp[5], Jan-Peter Grunz [6], Henner Huflage[6],
Rafael G. Jakubietz[1], Süleyman Ergün [7] & Rainer H. Meffert[1]

## Abstract

**Background** Traumatic separation of the pubic symphysis can destabilize the pelvis and require surgical fixation to reduce symphyseal gapping. The traditional approach involves open reduction and the implantation of a steel symphyseal plate (SP) on the pubic bone to hold the reposition. Despite its widespread use, SP-fixation is often associated with implant failure caused by screw loosening or breakage.

**Methods** To address the need for a more reliable surgical intervention, we developed and tested two titanium cable-clamp implants. The cable served as tensioning device while the clamp secured the cable to the bone. The first implant design included a steel cable anterior to the pubic symphysis to simplify its placement outside the pelvis, and the second design included a cable encircling the pubic symphysis to stabilize the anterior pelvic ring. Using highly reproducible synthetic bone models and a limited number of cadaver specimens, we performed a comprehensive biomechanical study of implant stability and evaluated surgical feasibility.

**Results** We were able to demonstrate that the cable-clamp implants provide stability equivalent to that of a traditional SP-fixation but without the same risks of implant failure. We also provide detailed ex vivo evaluations of the safety and feasibility of a trans-obturator surgical approach required for those kind of fixation.

**Conclusion** We propose that the developed cable-clamp fixation devices may be of clinical value in treating pubic symphysis separation.

### Plain language summary

Life-threatening pelvic injuries are often associated with disruption of a joint within the hip bones, called the pubic symphysis. Disruption can lead to a gap and subsequent instability of the pelvis. The current treatment is to stabilize the joint with a steel plate and screws, however this often becomes unstable soon after the operation. In this study, we analyzed two alternatives for stabilization that use cables and clamps instead of the plate. Further, we tested a surgical approach for implantation. The cables and clamps were as stable as a steel plate so offer an alternative approach to stabilize the pubic symphysis.

[1] Department of Orthopaedic Trauma, Hand, Plastic and Reconstructive Surgery, University Hospital Würzburg, Oberdürrbacher Str. 6, 97080 Würzburg, Germany. [2] Headmade Materials, Langhausstraße 9, 97294 Unterpleichfeld, Germany. [3] Center of Musculoskeletal Medicine, University Hospital LMU Munich, Marchioninistr. 15, 81377 Munich, Germany. [4] Department of Urology, University Hospital Würzburg, Oberdürrbacher Str. 6, 97080 Würzburg, Germany. [5] Additive Manufacturing Research Unit, SKZ Technology Center, Friedrich-Bergius-Ring 22, 97076 Würzburg, Germany. [6] Department of Radiology, University Hospital Würzburg, Oberdürrbacher Str. 6, 97080 Würzburg, Germany. [7] Institute of Anatomy, Julius-Maximilians-University Würzburg, Koellikerstraße 6, 97070 Würzburg, Germany. ✉email: Jordan_M@ukw.de

**Symphyseal disruption and plate fixation**. Rupture of the pubic symphysis is a traumatic pelvic injury in which the strong midline union of the pubic bones tears apart, requiring surgical fixation whenever the stability of the pelvis is impaired. Currently, the preferred treatment option is open reduction and symphyseal plate (SP) fixation using a steel plate and screws[1–3]. Despite the technique being used for decades in orthopedic traumatology, there is a severe hazard of SP implant failure and recurrent gapping. Loosening and breakage are some of the common complications (Supplementary Fig. 1). Some patients present only with mild radiologically verifiable plate loosening with an acceptable functional outcome (30–75%), whereas others require surgical revision and have a poorer outcome (3–11%)[4–7]. Reasons for the high failure rate are controversially discussed. Some experts suspect wrong indications rather than deficits of the implant. However there is progressive data showing that SP fixation is disadvantageous because it does not adequately neutralize the forces on the pubic symphysis during mobilization of the patient throughout the healing phase[8,9].

**Biomechanical considerations of symphyseal plating**. Previous biomechanical studies confirmed that SP fixation is unable to achieve stability similar to the uninjured pubic symphysis[8–10]. In young patients with good bone quality SP fixation can secure stability for several weeks until fibrocartilage healing of the pubic symphysis sets in. Whenever screw anchoring is impaired, for instance in osteoporotic bone, risk of implant failure rises before healing is completed[11,12]. Therefore, SP fixation might not be the ideal treatment option for all patients. The ongoing search for alternative techniques underscores the controversy surrounding SP fixation. Suture tapes[13,14], percutaneous screws[15], and the use of spine implants for internal fixation[16,17] are just some of the recently introduced alternative fixation techniques. The current literature confirms that there is a need for an alternative fixation device without the risk of screw loosening or plate breakage.

**Hypothesis of an alternative fixation technique**. Early reports about wire fixation of the pubic symphysis exist, a technique without the risk of screw loosening or breakage[18,19]. Fixation in which a wire is passed through the adjacent bone of the pubic symphysis or through the obturator foramen are reported. The limited tensile strength of wires is one of the reasons why such wire fixation is outdated in the meantime. Better fixation strength of braided steel cables renewed the discussion about cable fixation of the pubic symphysis[20,21]. However, a cable alone is at risk for dislocation and loosening because it lies solely on the bony surface without osseous fixation[20]. Therefore, an anchoring device for the cable seems to be required. The purpose of this study was to develop different anchoring devices that allow a cable as tensioning element to stabilize the pubic symphysis. The two anchoring devices tested here differ in their methods of stabilizing the pubic symphysis and were therefore separately evaluated. The cable-clamp anterior only (CCAO) design located the cable only on one side of the pubic symphysis to simplify cable placement outside the pelvis. The cable-clamp anterior and posterior (CCAP) design stabilized the anterior pelvic ring using a cable encircling the pubic symphysis to provide a potentially higher stability than that supplied by the CCAO design.

We hypothesized that cable-clamp fixation would be a possible alternative to SP fixation. We conducted an equivalence and non-inferiority testing to proof effectiveness of the developed implant devices compared to SP-fixation but with better properties such as fewer risk of implant failure or less invasive implantation[22]. To obtain sufficient data for statistical analysis of our approach, we first used a comprehensive synthetic bone model because it is

available in unlimited quantities and is highly reproducible. Next, we conducted a smaller biomechanical cadaver study to simulate conditions in vivo. Although cadaver specimens are difficult to obtain and have a greater variability, combining both methods in a two-phase testing regime provides a more complete biomechanical understanding. Further, a cadaver study was conducted to evaluate the safety and feasibility of the trans-obturator surgical implantation. We were able to confirm equivalent stability of SP fixation and both cable-clamp prototypes.

## Materials and methods

**Implant manufacturing**. To address the need for a more reliable surgical intervention, we developed and tested two titanium cable-clamp implants. The implant system consisted of a tensible steel cable and two clamps, each clamp possessing a cable-guiding channel and a large opening to attach the clamp to the bone using a screw. The two implant designs differed in the path of their guiding channels, with the CCAO design having the guiding channel on only one side of the clamp and the CCAP design having the guiding channel running along both sides. The purpose of the clamp was to attach the cable to the bone, securely guide the cable close to the bone, increase the contact area of the cable and the bone, and avoid cable loosening. Clamps were planned with mirror symmetry and a clockwise thread. After trialing different prototypes, we fabricated slim, anatomically contoured clamps (Fig. 1) by combining selective laser sintering with a new metal/binder powder feedstock and industrial processes such as conventional sintering, a technique called cold-metal-fusion 3D printing (CMF). Commercially available titanium powder (Ti6Al4V) was suited to this purpose and integrated into the binder system as required. After a processable feedstock was established, test specimens were produced to assess their density and chemical composition. Because the clamp overhang depended on the bone geometry, incorporation of support structures within the design was trialed. Afterward, test clamps were printed using selective laser sintering (Formiga P110, EOS, Germany). The binder was then dissolved using furnace heating, leaving a solid metal implant. Compensation for shrinkage during this process was calculated in advance (Supplementary Fig. 2)[23,24].

**Biomechanical testing of synthetic bone models**. Thirty synthetic pelves (Pelvis Complete, Art. No. 4060, Synbone, Zizers, Switzerland) were used in our biomechanical study. We justify the use of synthetic bone models with its widely accepted use in biomechanical laboratories and its recent utilization in similar pelvic studies allowing comparability of our results[13,25–28]. An anterior–posterior compression injury (Young and Burgess APC II; OTA/AO: 61-B2.3d) was simulated[29,30], leading to disruption of the pubic symphysis and the iliosacral connections. Fixation of posterior instability was conducted in all specimens using a partially threaded stainless steel screw (ø 6.5 mm, 95 mm long) on the right sacroiliac joint. The disrupted synthetic bone models were divided into three groups. The control group underwent SP fixation (Group SP; $N = 10$) using a four-hole stainless steel symphysis plate (Symphyseal Plate 3.5 Synthes, Oberdorf, Switzerland; Art. No. 02.100.004) and four self-tapping, non-locking cortex screws (ø 3.5 mm, Synthes, Oberdorf, Switzerland). Medial and lateral screws were 55 mm and 60 mm long, respectively. Attention was paid to the length and convergence angle of the screws to avoid their penetration through the cortical bone. Pubic symphysis disruption in Group CCAO ($N = 10$) was fixed using the CCAO device by securing each V-shaped clamp to the cranio-medial edge of opposing obturator foramina with a fully threaded stainless steel screw (M4; ø 4 mm, 20 mm long). After fixation,

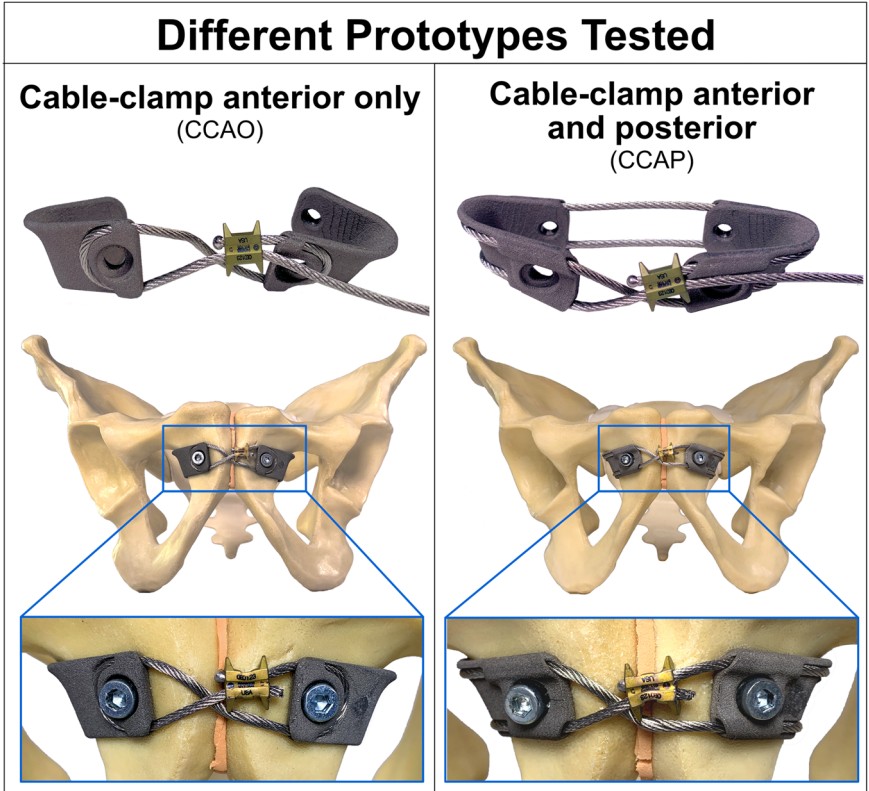

**Fig. 1 Cable-clamp fixation devices for stabilization of a ruptured pubic symphysis.** The cable-clamp anterior only (CCAO) design positions the cable anterior to the symphysis. The cable-clamp anterior and posterior (CCAP) design requires the cable to encircle the pubic symphysis. Both clamps contain cable-guiding surface channels and large openings for screw placement.

the pubic rami were reduced using a pointed reduction clamp, then a steel braided cerclage cable (∅ 1.7 mm, Synthes, Oberdorf, Switzerland) was pulled through the anterior guiding channels of the clamps. Finally, the centrally crossed cerclage cable was pretensioned using a cable tensioning device to reduce the symphysis, with care taken to ensure that the tension forces were equal. After anatomical reduction, the cramp was deformed and locked using a ratcheting cable crimper and shortened to a minimum using a cable cutter. Pubic symphysis disruption in Group CCAP ($N = 10$) was fixed using the CCAP device by looping the cable through the obturator foramina, following the guiding channels of the clamps, then tensioned, secured, and shortened. Thus, the symphysis was looped with the cable cerclage twice. All specimens were examined using X-ray imaging to verify similar implant positioning (Fig. 2).

For biomechanical testing setup we used a single-leg-stance model as it is simple, has minimal risk for confounders, and it is well established by other groups[31,32]. Hereby vertical sheer stress can be applied to the pubic symphysis but no traction. Different biomechanical test set-ups are mentioned in the literature, like the two-leg-stance with simultaneous load or with alternating load, but all have separate limitations[9,33]. The pelvic models were fixed to the material testing machine (Z020, Zwick/Roell GmbH, Ulm, Germany) using a custom-made aluminum device connected to a force transducer. Pelvic samples were attached via the sacrum at a physiological 45° tilt. The acetabulum articulated with a hemiarthroplasty femoral head prosthesis (∅ 48 mm), and the physiological antetorsion angle of 15° was taken into account. In addition, a hook system was attached to the right ileum to simulate the musculus tensor fasciae latae and the tractus iliotibialis. A series of pre-tests were performed, using up to 12,000 test cycles and load levels up to 600 N, until a final test

protocol was defined (Table 1A). The testXpert II software (Version 3.6, Zwick/Roell GmbH) continuously recorded force and displacement to generate a load–displacement curve and measure the following parameters: peak-to-peak displacement (mm), total displacement (mm), plastic deformation (mm), and stiffness (N/mm). A coordinate system was prepared by attaching round visual markers (1.5 mm i.d., 3.5 mm o.d.) in one plane, and an optical measuring system (ARAMIS 3D Professional, Carl Zeiss GOM Metrology GmbH, Braunschweig, Germany) equipped with two 6 MP cameras/a 12 MP sensor (GOM) and ARAMIS Professional software 2018/2019 (Carl Zeiss GOM Metrology GmbH, Braunschweig, Germany) was linked to the material testing machine and used to follow the movement of the markers.The visual markers served as identifiers at different anatomical landmarks for the 12 megapixel cameras/sensor and a reference block (140 × 22 × 32 mm) with three point markers was defined as coordinate system as a reference. This enabled the displacement of the individual visual markers to be measured in X, Y, and Z axis. For measurement, the 30 visual measuring points were divided into the following point components: coordinate system, femur, right symphysis, left symphysis, right sacroiliac joint, and left sacroiliac joint (Supplementary Fig. 3). Calibration was performed before each test and accuracy was confirmed (0.3–0.03 mm). Among the several visual measuring points, we determined those on either side of the symphysis to be most relevant, and we referred to it as symphyseal displacement (mm). This parameter allows to indirect measure movement of the pelvic ring. The optical measuring system was adjusted and coupled to the material testing machine to record the maximum loads of the 10 setting cycles during pre-testing, the first measuring cycle, and every 20th measuring cycle up to the 5000th (Fig. 3).

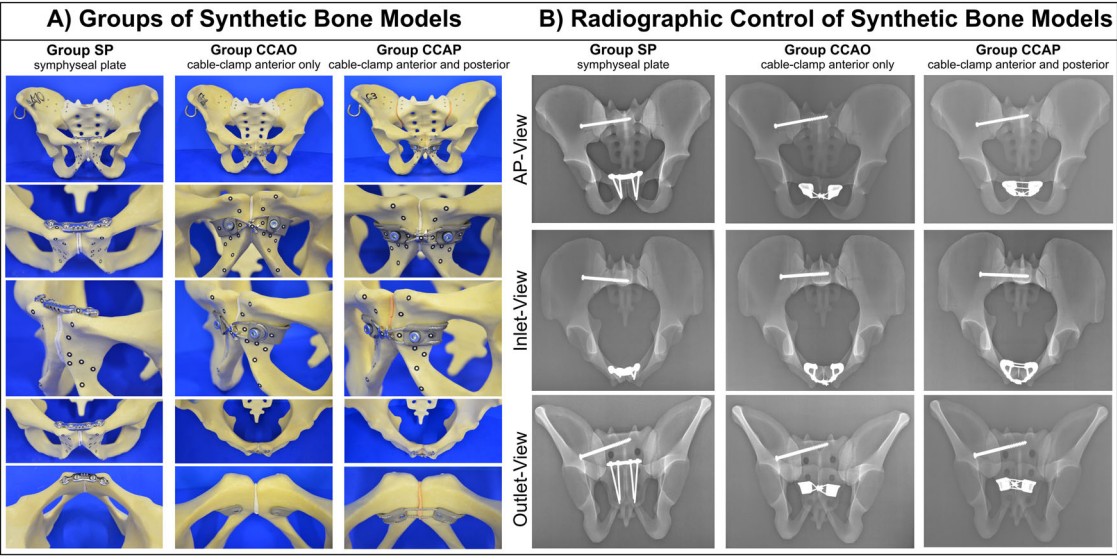

**Fig. 2 Biomechanical test specimens. A** Photographs of SP, CCAO, and CCAP devices attached to synthetic bone models used for biomechanical testing. The gold-standard four-hole SP served as a control. The CCAO device uses an extrapelvic cable, whereas the CCAP device uses a cable that enters the obturator foramen and surrounds the pubic symphysis. **B** X-ray images of SP, CCAO, and CCAP devices attached to synthetic bone models, shown from anterior–posterior (AP), inlet, and outlet views to confirm correct implant positioning.

**Table 1 A) Material testing protocol used to examine synthetic pelvis models. B) Material testing protocol used to examine cadaver pelves.**

| | Cycles | Maximum | Minimum | Frequency |
|---|---|---|---|---|
| **A) Synthetic bone** | | | | |
| 1. Setting cycles | 10 | 50 N | 10 N | 50 mm/min |
| 2. Measuring cycles | 5000 | 300 N | 50 N | 100 mm/min |
| 3. Setting cycles | 10 | 50 N | 10 N | 50 mm/min |
| 4. Measuring cycles | 5000 | 400 N | 50 N | 100 mm/min |
| **B) Cadaver bone** | | | | |
| 1. Setting cycles | 10 | 50 N | 10 N | 50 mm/min |
| 2. Measuring cycles | 1000 | 300 N | 50 N | 100 mm/min |

**Biomechanical testing of cadaver bone models**. No approval of the ethics committee of the University of Würzburg was necessary for the description of the surgical technique and biomechanical testing outlined in this article. Informed consent to participate in research questions was given and all cadavers were anonymized for this project. Six fresh-frozen cadaver pelvic bones were used for biomechanical analysis (3 male, 3 female). Extirpation of the full pelvis was carried out, and all muscular attachments were removed. The ligamentum sactrotuberale and ligamentum sacrospinale were dissected on one side of the pelvis and retained on the other. The pubic symphysis and the anterior sacroiliac ligament were incised to create an anterior–posterior compression type II injury. Three specimens were stabilized using the CCAO fixation device and three were stabilized using the CCAP device. The human pelvis was attached to the biomechanical testing setup in a manner similar to that of the synthetic bone models but without hook attachment to the right ileum, and biomechanical testing was performed on the same day as fixation. In this special setting we decided against the ileum attachment to observe movement at the anterior pelvic ring without limitation. The optical measuring system was activated, and the test protocol was adapted for cadaver specimens (Table 1B).

**Establishment of a trans-obturator approach using a cadaver model**. Six fresh-frozen cadaver specimens underwent the following sequence of surgical steps: incision of the symphysis pubis, implantation of the medical device (Supplementary Videos 1–4), soft tissue closure, computed tomography (CT) scanning (bladder filled with radiocontrast fluid), full anatomical dissection, and final extirpation of the pelvis. State-of-the-art twin-beam CT (SOMATOM Definition Edge; Siemens Healthcare GmbH, Erlangen, Germany) or photon-counting CT (NAEOTOM Alpha; Siemens Healthcare GmbH) scanners were used for cadaveric pelvis imaging (Table 2). Scan parameters were chosen to optimize image quality. Prefiltration techniques (Au+Sn 120 kV/Sn 140 kV) were used to minimize streak artifacts caused by the metal implants. In addition to conventional CT, cystograms were acquired using a standard Foley catheter. Therefore, iodine contrast agent (Imeron 350; Bracco Imaging Deutschland GmbH, Germany) diluted with water (1:5) was injected into the urinary bladder to rule out bladder or urethra laceration. Irrespective of the scanner used, reconstructions in axial, coronal, and sagittal orientations were performed employing a sharp bone kernel. The slice thickness and increment were chosen with 1 mm and 0.5 mm, respectively. Additionally, cinematic volume rendering was performed using dedicated software (syngo.via VB60, Siemens Healthcare GmbH).

**Statistics and reproducibility**. A power analysis was performed using a power of 80% and a significance level of 5% to confirm that the sample size for synthetic bone models was adequate. The results are presented as mean values with standard deviation. All data underwent statistical analysis for normal distribution using the Shapiro–Wilk test. Analysis of variance was used to compare the means. A post hoc test was not necessary because no difference was measurable. A $P$ value of less than 0.05 was considered statistically significant. Further, data acquired using the synthetic bone model underwent equivalence and non-inferiority testing. Use of confidence intervals is preferred for statistical testing of non-inferiority (i.e., equivalence or superiority). Calculation of confidence intervals requires data to have a normal distribution

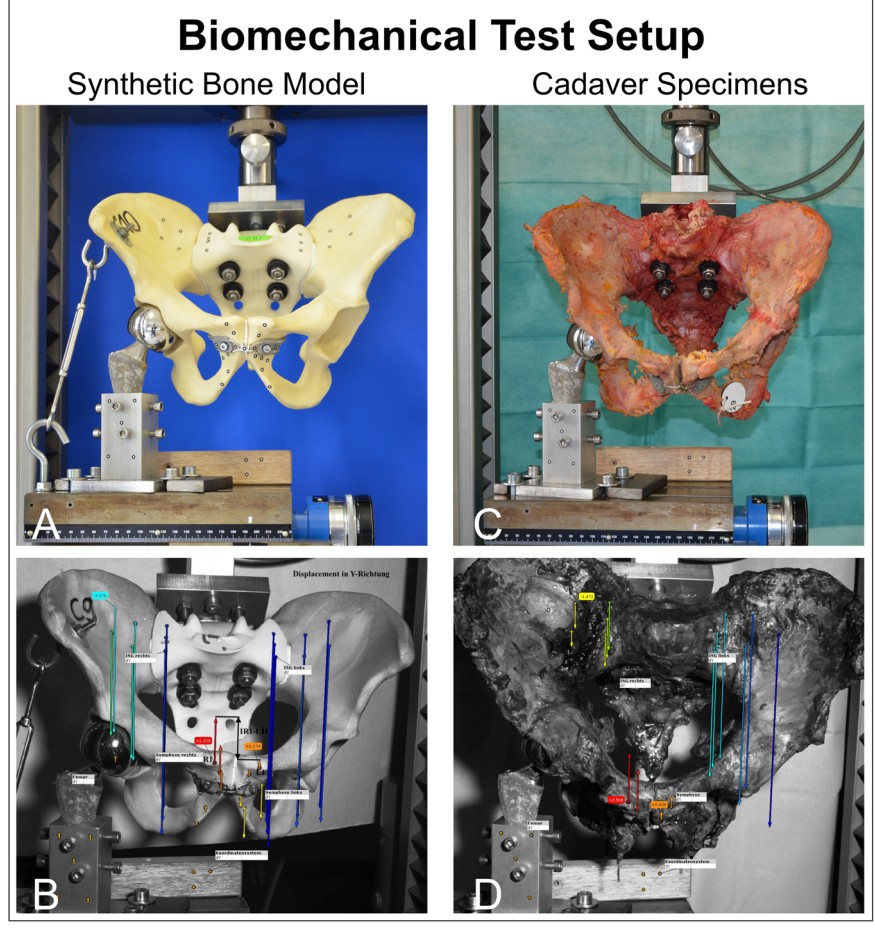

**Fig. 3 Biomechanical testing setup. A** A synthetic bone model, fixed with one of the developed implants, was mounted to the material testing machine. A single-leg stance test with uniaxial load was used. **B** Visual markers were placed around the pubic symphysis and iliosacral joints and their motion was tracked using a camera system. The material testing machine measured the displacement of the whole pelvis, whereas optical video analysis was used to measure bone movement at the pubis symphysis. The methods of attachment (**C**) and optical measurement (**D**) of cadaver bones were similar to those used for the synthetic bone models. Colored arrows in panels (**B** and **D**) indicate movements along the y-axis. The number of cycles used to test cadaver bones was lower than that used for testing the synthetic bone models because of the limited time available after cadaver specimens were defrosted.

**Table 2 Synopsis of the cadaver specimens.**

| Nr. | Gender | Implant | Approach | Remarks | Scanner |
|---|---|---|---|---|---|
| 1 | Male | CCAO | Trans-obturator horizontal incision | Implant well aligned; bladder intact; prostate carcinoma; bone metastasis. | SOMATOM Definition Edge; Siemens Healthcare GmbH |
| 2 | Female | CCAP | Trans-obturator horizontal incision | Cable at one posterior site not within the guiding channel; bladder intact; morbidly obese. | SOMATOM Definition Edge; Siemens Healthcare GmbH |
| 3 | Male | CCAO | Trans-obturator horizontal incision | Implant well aligned; bladder intact. | SOMATOM Definition Edge; Siemens Healthcare GmbH |
| 4 | Female | CCAP | Trans-obturator horizontal incision | Anterior bladder laceration; leakage of contrast fluid around the pubic symphysis; osteopenia; fracture adjacent to the pubic symphysis. | SOMATOM Definition Edge; Siemens Healthcare GmbH |
| 5 | Male | CCAO | Trans-obturator midline incision | Implant well aligned; bladder intact. | NAEOTOM Alpha; Siemens Healthcare GmbH |
| 6 | Female | CCAP | Trans-obturator midline incision | Implant well aligned; bladder intact. | NAEOTOM Alpha; Siemens Healthcare GmbH |

*CCAO* cable-clamp anterior only, *CCAP* cable-clamp anterior and posterior.

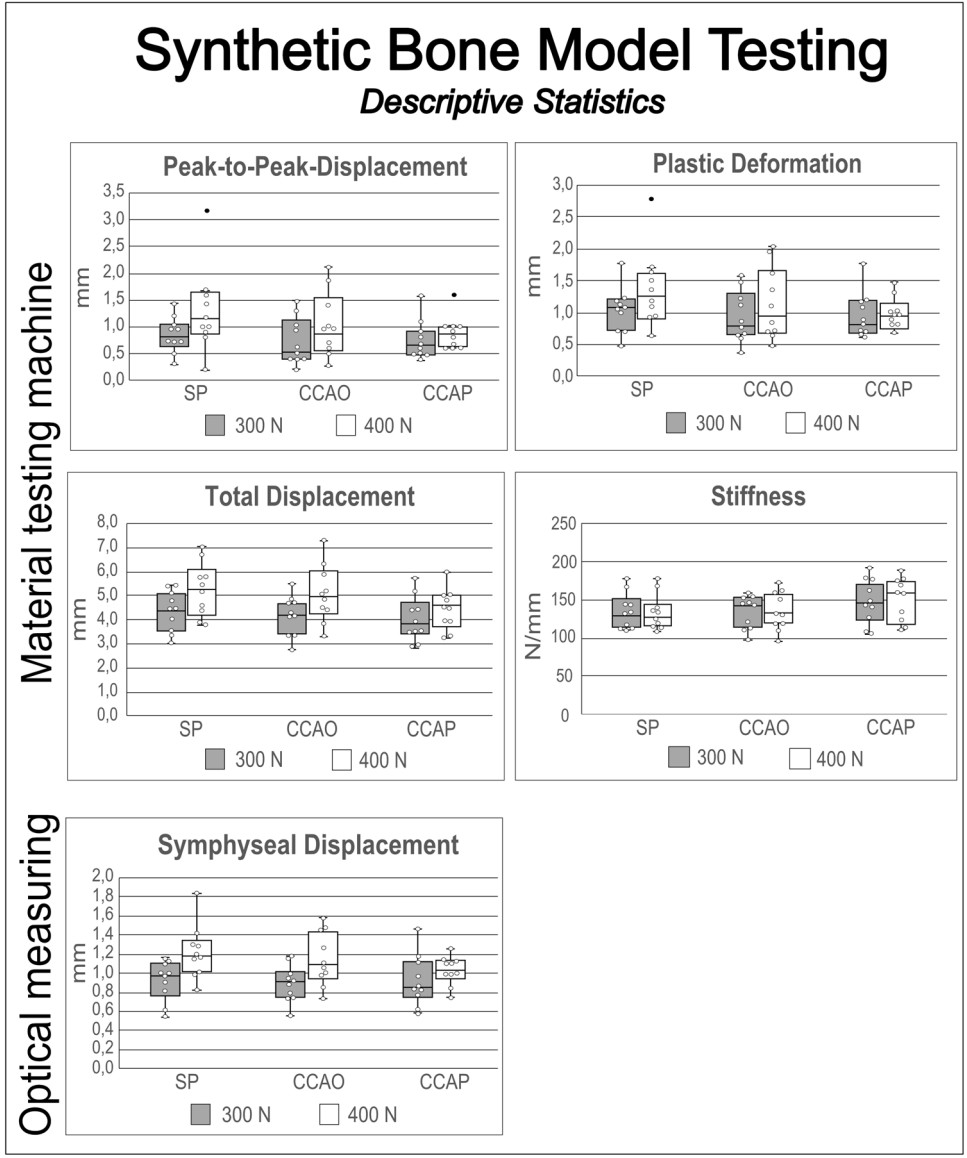

**Fig. 4 Results of the synthetic bone model biomechanical testing - box plots (*N* = 10).** The material testing machine generated a load–displacement curve from which peak-to-peak displacement, plastic deformation, total displacement, and stiffness were measured under two different cyclic load levels. The optical system measured changes in separation of the pubic branches (symphyseal displacement). White dots = individual data points. Black dots = outlier. Lines in the box and whisker plot = error bars are the 95% confidence interval, the bottom and top of the box are the 25th and 75th percentiles, and the line inside the box is the 50th percentile (median).

and homogenous variance, which we assessed using the Shapiro-Wilk and Levene tests, respectively. When necessary, a square root transformation of the data was performed to ensure that test requirements were satisfied. All data generated or analysed during this study are included in this published article (Supplementary Data 1).

**Reporting summary**. Further information on research design is available in the Nature Portfolio Reporting Summary linked to this article.

## Results

**Implant manufacturing and handling**. Solid titanium implants with CCAO and CCAP designs were manufactured using CMF. The thread of the opening for screw insertion was hand-cut to improve accuracy. Biomechanical testing showed that the CCAO device was easier to handle than the CCAP device. The extra-pelvic cable in the CCAO device ran smoothly along the guiding structure on the surface of the clamp, thereby reducing the risk of irritating surrounding anatomical structures, and the intrapelvic part of the implant was small and fully covered by soft tissue in the cadaver specimens. Cable placement in the CCAP device was slightly more cumbersome. In contrast to the synthetic bone model, the cable could only pass once around the pubic symphysis in the cadaver because limited visual access made threading it through the posterior cable guide of the in situ implant difficult (Supplementary Videos 1 and 2). Vigorous use of an oversized cable passer bears the risk of bladder laceration as observed in one case. In a gapping traumatic symphyseal rupture, retraction of the bladder might enable the cable to be more safely passed. We decided to perform an additional cranial midline incision between the pyramidalis muscles to improve visualization (Supplementary Videos 3 and 4). Provided one is aware of

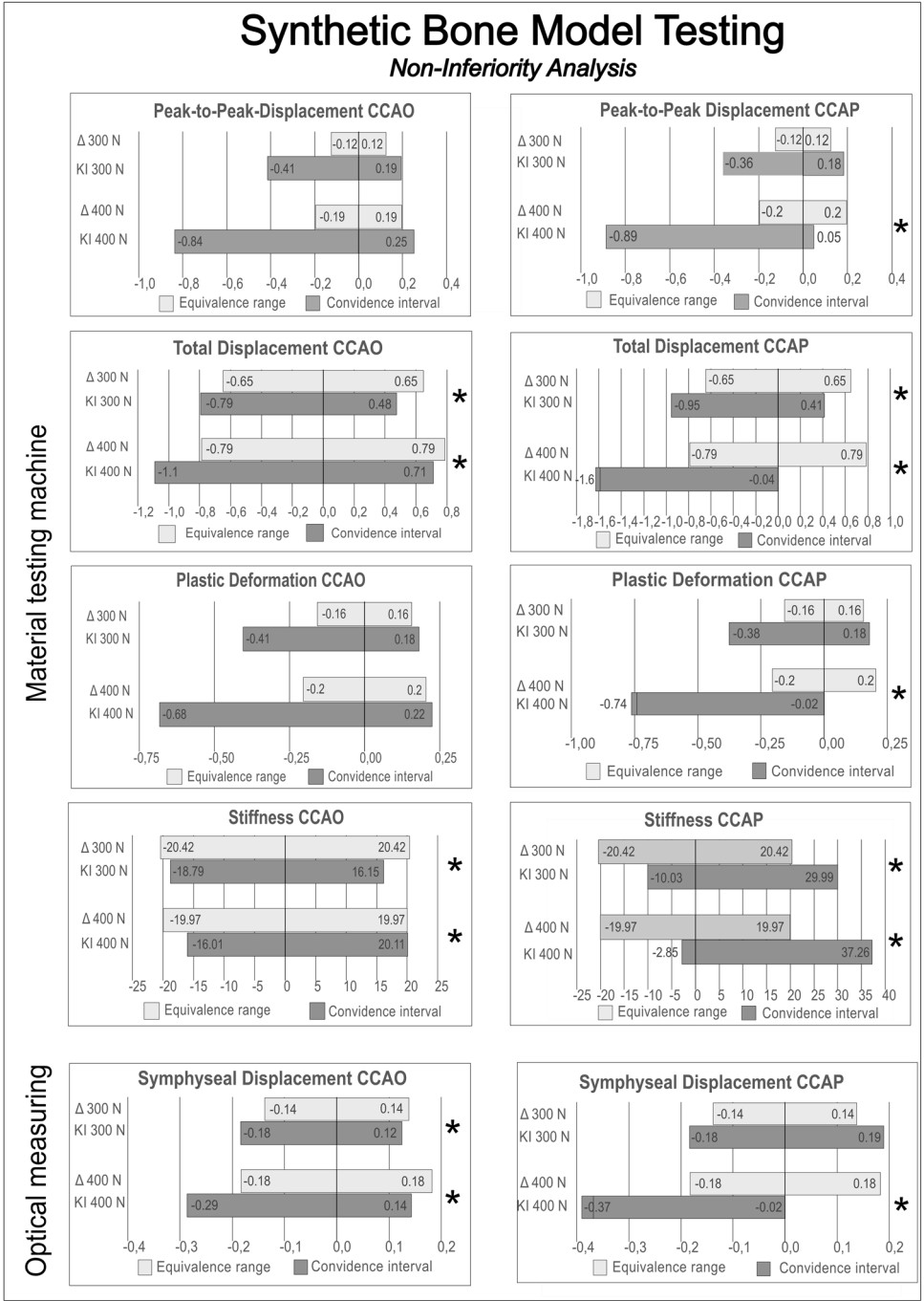

**Fig. 5 Results of the synthetic bone model biomechanical testing - non-inferiority testing.** Non-inferiority testing of the synthetic bone models confirmed equivalence of SP and cable-clamp (CCAO or CCAP) fixation methods for several of the test parameters (asterisk). Absence of non-inferiority does not indicate lower stability because the non-inferiority test could not be conducted for some parameters.

this problem, we assume that the risk of bladder laceration is manageable. Nevertheless, bladder laceration is a hazard and is not specific to the implant presented here[34,35].

**Biomechanical analysis of the synthetic pelvic bone model.** The peak-to-peak displacements of the entire pelvic ring were 0.83 ± 0.33 mm (SP), 0.72 ± 0.44 mm (CCAO), and 0.74 ± 0.37 mm (CCAP) under 300 N load and 1.31 ± 0.80 mm (SP), 1.02 ± 0.59 mm (CCAO), and 0.89 ± 0.30 mm (CCAP) under 400 N. Non-inferiority of the CCAO device compared with the SP device could not be proved under either load. Non-inferiority of

the CCAP device compared with the SP device could be proved under 400 N load but not 300 N (Fig. 4 & 5; Table 3).

The total displacements of the pelvic ring were 4.31 ± 0.83 mm (SP), 4.15 ± 0.81 mm (CCAO), and 4.04 ± 0.93 mm (CCAP) under 300 N load and 5.25 ± 1.14 mm (SP), 5.06 ± 1.19 mm (CCAO), and 4.44 ± 0.85 mm (CCAP) under 400 N. Non-inferiority of both the CCAO and CCAP devices compared with the SP device was proved under both loading conditions.

Plastic (irreversible) deformation of the whole pelvic ring was determined using the load–displacement curve and the measured values were 1.05 ± 0.35 mm (SP), 0.94 ± 0.40 mm (CCAO), and 0.95 ± 0.36 mm (CCAP) under 300 N load and 1.36 ± 0.60 mm

**Table 3 Statistical analysis of the synthetic bone models tested by ANOVA.**

**Multivariate ANOVA for synthetic bone models**

| Parameter | Group | Mean | STD | P-value |
|---|---|---|---|---|
| Stiffness [N /mm] 300 N | SP | 136.2 | 23.6 | 0.535 |
| | CCAO | 134.8 | 21.5 | |
| | CCAP | 146.1 | 27.9 | |
| Stiffness [N /mm] 400 N | SP | 133.2 | 22.5 | 0.268 |
| | CCAO | 135.2 | 24.0 | |
| | CCAP | 150.4 | 28.8 | |
| Peak-to-Peak-Displacement [mm] 300 N | SP | 0.832 | 0.333 | 0.795 |
| | CCAO | 0.722 | 0.442 | |
| | CCAP | 0.744 | 0.366 | |
| Peak-to-Peak-Displacement [mm] 400 N* | SP | 1.309 | 0.799 | 0.292 |
| | CCAO | 1.018 | 0.591 | |
| | CCAP | 0.890 | 0.299 | |
| Total Displacement [mm] 300 N | SP | 4.312 | 0.826 | 0.781 |
| | CCAO | 4.153 | 0.809 | |
| | CCAP | 4.042 | 0.933 | |
| Total Displacement [mm] 400 N | SP | 5.252 | 1.137 | 0.222 |
| | CCAO | 5.064 | 1.187 | |
| | CCAP | 4.438 | 0.849 | |
| Plastic Deformation [mm] 300 N | SP | 1.050 | 0.354 | 0.768 |
| | CCAO | 0.938 | 0.402 | |
| | CCAP | 0.951 | 0.362 | |
| Plastic Deformation [mm] 400 N | SP | 1.360 | 0.603 | 0.248 |
| | CCAO | 1.132 | 0.564 | |
| | CCAP | 0.979 | 0.263 | |
| Symphyseal Displacement 300 N | SP | 0.919 | 0.210 | 0.939 |
| | CCAO | 0.890 | 0.188 | |
| | CCAP | 0.922 | 0.267 | |
| Symphyseal Displacement 400 N | SP | 1.218 | 0.276 | 0.210 |
| | CCAO | 1.146 | 0.278 | |
| | CCAP | 1.023 | 0.152 | |

The results are presented as mean values with standard deviation. All data underwent statistical analysis for normal distribution using the Shapiro–Wilk test (*no normal distribution). Analysis of variance was used to compare the means. A post hoc test was not necessary because no difference was measurable. A P value of less than 0.05 was considered statistically significant. Absence of a significant difference cannot be interpretated as equivalence of the implants. Therefore, the non-inferiority testing was necessary.

(SP), $1.13 \pm 0.56$ mm (CCAO), and $0.98 \pm 0.26$ mm (CCAP) under 400 N. Non-inferiority of the CCAO device compared with the SP device could not be proved under either loading condition. Non-inferiority of the CCAP device compared with the SP device could be proved under 300 N load but not 400 N.

Stiffness values were determined from the slope of the load–displacement curve during the fifth cycle and measured to be $136.15 \pm 23.55$ N/mm (SP), $134.83 \pm 21.47$ N/mm (CCAO), and $146.13 \pm 27.87$ N/mm (CCAP) under 300 N load and $133.15 \pm 22.53$ N/mm (SP), $135.21 \pm 24.02$ N/mm (CCAO), and $150.36 \pm 28.81$ N/mm (CCAP) under 400 N. Non-inferiority of both CCAO and CCAP devices compared with the SP device was proved under both loading conditions.

Symphyseal displacements were determined using 3D optical analysis and measured to be $0.92 \pm 0.21$ mm (SP), $0.89 \pm 0.19$ mm (CCAO), and $0.92 \pm 0.27$ mm (CCAP) under 300 N load and $1.22 \pm 0.28$ mm (SP), $1.15 \pm 0.28$ mm (CCAO), and $1.02 \pm 0.15$ mm (CCAP) under 400 N. Non-inferiority of the CCAO device compared with the SP device was proved under both loading conditions, whereas the CCAP device was non-inferior only under the load of 400 N.

**Biomechanical analysis of the cadaver pelvic bone model.** All cadaver tests were conducted under a load of 300 N (Table 1B). The peak-to-peak total pelvic displacements were $2.62 \pm 2.18$ mm (CCAO) and $2.72 \pm 1.26$ mm (CCAP). Total displacements were $9.50 \pm 0.87$ mm (CCAO) and $4.94 \pm 1.26$ mm (CCAP). Plastic deformations were $3.05 \pm 1.85$ mm (CCAO) and $3.27 \pm 1.47$ mm (CCAP). Stiffness values were $191.59 \pm 62.45$ N (CCAO) and $119.87$ N $\pm 53.54$ (CCAP). The mean symphyseal displacements were $1.29 \pm 0.89$ mm (CCAO) and $1.83 \pm 0.70$ mm (CCAP). Non-inferiority testing was not conducted for the cadaver specimens because of the small sample size (Fig. 6).

**Establishment of a trans-obturator approach using a cadaver model.** The cadavers were positioned in a supine position on a flat table, and a Foley catheter was inserted. A midline or horizontal incision (8–10 cm long) was made directly anterior to the pubic symphysis. During preparation through the skin and soft tissue to the pubic symphysis, the spermatic cord or round ligament were spared. Subsequently, the pubic symphysis was dissected. To expose the medial border of the obturator foramen, the pectineus and external obturator muscle were partially detached. Trans-obturator access to the pelvic cavity was gained by careful digital dissection though the obturator membrane and the internal obturator muscle including the fascia. The puborectalis muscle was also pierced. This procedure was performed on both sides of the pelvis to enable the fingertips to access the posterior part of the pubic symphysis. Visual access was limited because of the intact symphysis. The clamps of the implant were manually placed at the medial border of the obturator foramen under visual control. Intrapelvic palpation using the index finger confirmed no incarceration of the bladder. A 1.7 mm braided steel cable was passed through the guiding channels of the clamps, which were secured to the pubis by drilling and screw fixation. Using the tensioning device, the cable was tightened to compress the pubic symphysis. The tension and position of the cable were secured by deformation of the clamp, as recommended by the manufacturer (Cable System, Surgical Technique, DePuy Synthes). The cable was shortened and tissue layers were closed above the symphysis (Supplementary Videos 1–4).

The surgical approach for implanting the CCAP device was different from that of the CCAO device to enable the cable to be passed around the posterior side of the pubic symphysis (Fig. 7). We additionally performed a 3 cm cranial extension next to the pubic symphysis along the midline of the pyramidalis muscles. When entering the pelvic cavity via the Retzius space, the bladder could be retracted and the cable passed safely. After wound closure, all specimens were evaluated using CT, and then? fully dissected. The following anatomical structures were at risk (Fig. 8): the artery, vein, and nerve inside the obturator canal; the spermatic cord or round ligament; the ilioinguinal nerve; the genitofemoral nerve (genital branch); the suspensory ligament of the penis; superficial dorsal vein of the penis; clitoral glans and body; the bladder and prostate; and the femoral artery, vein, and nerve. The full cadaver dissection showed a well-fitting implant sufficiently distanced from surrounding anatomical structures. The content of the muscular and vascular lacunae was laterally far from the operating area. The obturator nerve came closest to the implant. The ilioinguinal and the genitofemoral nerves had variable shapes and locations and could not be identified in all specimens. In male subjects, the spermatic cord was free from contact with the implant. In one cadaver, bladder laceration was identified (Fig. 9). Postoperative cadaver CT analysis confirmed a good implant fit to the pubic bone (Fig. 10). Both clamp types were located at the cranial–medial border of the obturator foramen, and there was sufficient distance to the obturator canal. There was no contact between the bladder (filled with iodine contrast agent) and the implant. Between the implant and the bladder was adipose tissue. The Foley catheter was clearly visible,

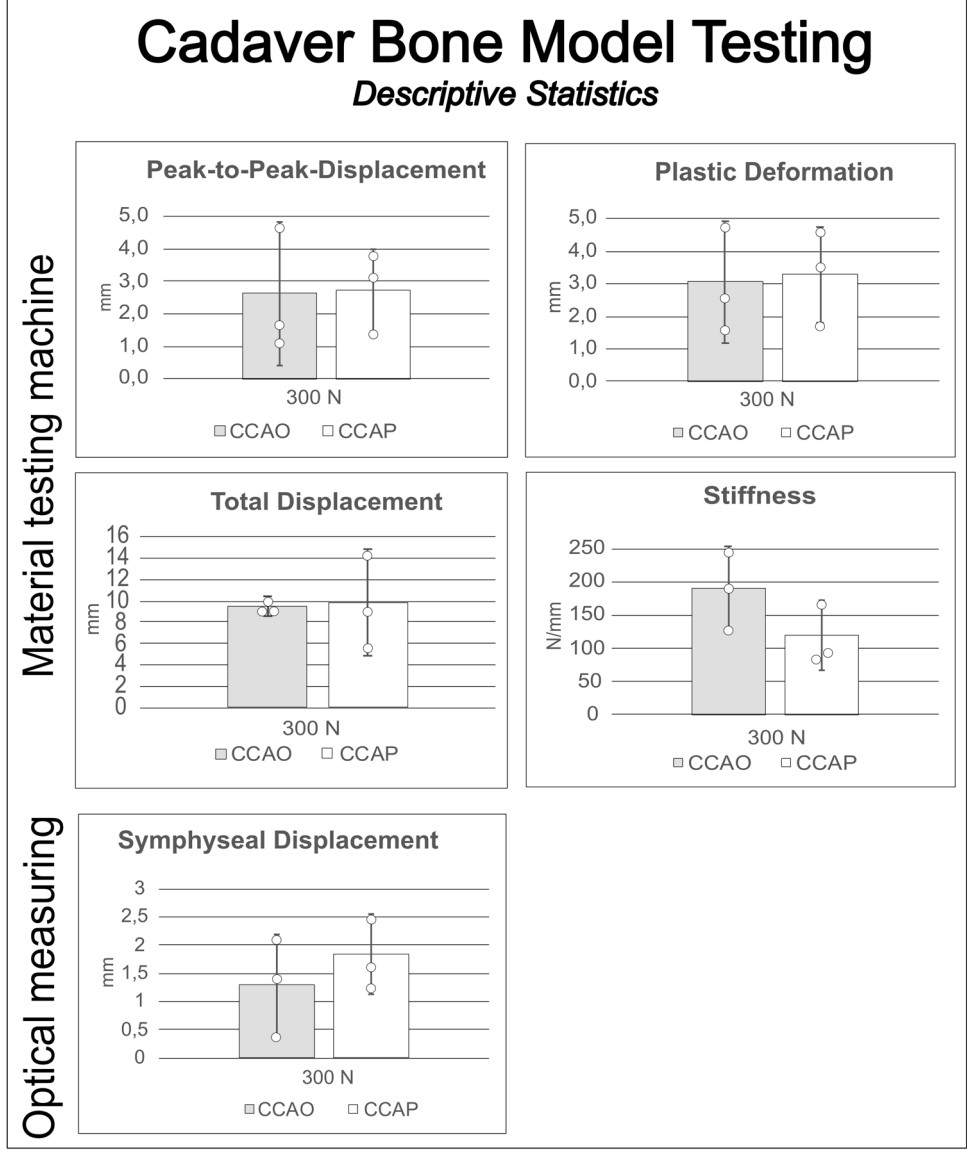

**Fig. 6 Results of the cadaver biomechanical testing (N = 3).** Parameters were similar to those obtained using the synthetic bone model. The material testing machine generated a load–displacement curve from which peak-to-peak displacement, plastic deformation, total displacement, and stiffness were measured under two different cyclic load levels. The optical system measured changes in separation of the pubic branches (symphyseal displacement). Sample sizes were smaller and SP fixation was not evaluated. White dots = individual data points.

and the urethra was free from contact with the implant in all cadavers. In male cadavers, there was no contact with the prostate (one specimen had prostatectomy). One female cadaver showed leakage of contrast fluid from the bladder, which was confirmed to be lacerated during the subsequent dissection. Bladder injury most likely occurred during the attempt to implant the CCAP device using a cable passer. Bone density varied between male and female subjects.

## Discussion
We attempted minimally invasive implantation of two medical devices for stabilization of a ruptured symphysis. Using a 3D printing technology, we fabricated solid and stable titanium implants with complex geometric structures.

Biomechanical analysis of synthetic bone models demonstrated that CCAO and CCAP designs were non-inferior to the regular SP fixation when comparing important parameters such as total displacement, symphyseal displacement, and stiffness. For some

other parameters, a non-inferiority test could not be conducted using the available data. Biomechanical analyses of cadavers confirmed the findings obtained using the synthetic bone models, but the results were more variable. We observed a trend for higher stability of the CCAP design compared with the CCAO design, but the difference was not statistically significant. Because we did not use a load-to-failure test, it remains to be determined whether the CCAP design has a higher ultimate strength. In our view, the combination of synthetic and cadaver bone models is essential for studying the biomechanics of symphyseal fixation devices. Synthetic bone models are readily available and highly reproducible but have limited capacity to mimic human musculoskeletal conditions[36]. Cadaver bone models are better able to mimic such musculoskeletal conditions but are usually highly variable because of differences among body donors[37]. Comparison of our results to the recent literature confirms a current pursuit of alternative fixation devices. Most recently Hinz et al. developed a flexible implant by combining plates with a fiber cord

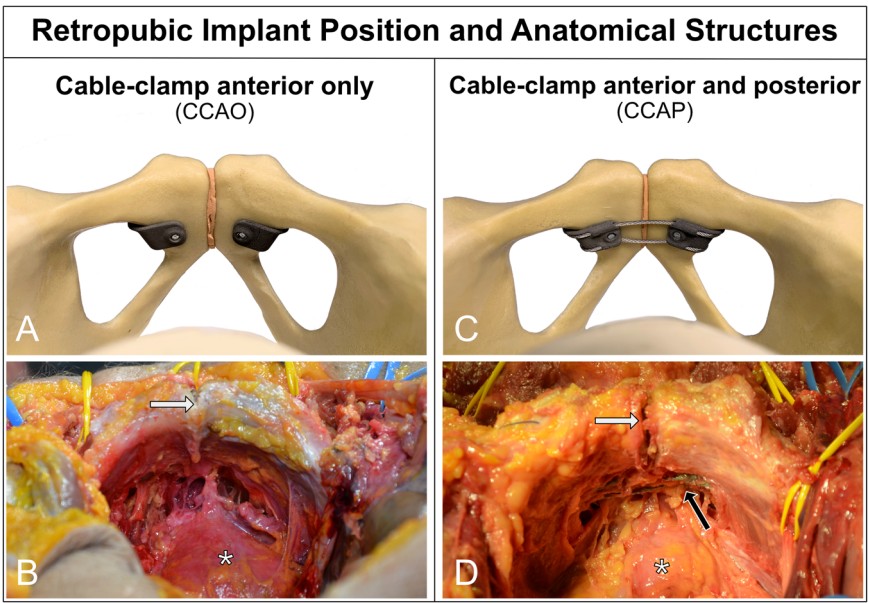

**Fig. 7 Retropubic implant position.** A slim implant design is desirable to avoid soft tissue irritation, especially in the retropubic space where only a slim soft tissue layer covers the implant and protects the bladder. Images show the position of the implant at the posterior border of the pubic bone that faces the bladder. **A** The CCAO device attached to a synthetic bone model. **B** The CCAO device implanted in a cadaver, where the implant is fully covered by soft tissue. **C** The CCAP device attached to a synthetic bone model, showing the cable than can run in a single or double loop (shown) around the pubic symphysis. **D** The CCAP device in a cadaver, showing the cable gleaming through the soft tissue in the space of Retzius (black arrow). In panels (**B** and **D**), the white arrow points towards the pubic symphysis and the asterisk marks the bladder.

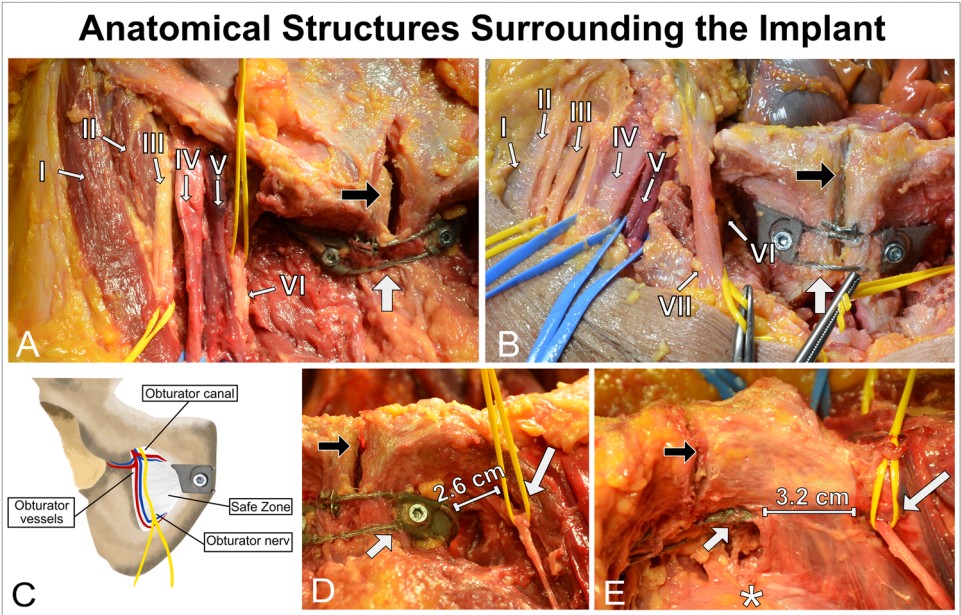

**Fig. 8 Anatomical structures surrounding the implant.** Cadaver study of the extrapelvic and intrapelvic area around the implant. **A** Anatomical structures of the muscular and vascular lacunae in a female cadaver. Identified are: (I) sartorius muscle, (II) iliopsoas muscle, (III) femoral nerve, (IV) femoral artery, (V) femoral vena, and (VI) obturator nerve. **B** Anatomical structures of the muscular and vascular lacunae in a male cadaver. Identified are: (I) sartorius muscle, (II) iliopsoas muscle, (III) femoral nerve, (IV) femoral artery, (V) femoral vena, (VI) obturator nerve, and (VII) spermatic cord. **C** Schematic illustration of the obturator foramen, showing the obturator canal where the obturator nerve and vessels pass out of the pelvis. The canal is located at the cranio-lateral border of the foramen, whereas the implant is on the opposite side. **D** Anterior view of the dissected obturator foramen. **E** Posterior oblique view of the obturator foramen covered by the fascia of the obturator internus muscle. An asterisk marks the bladder. Long white arrows indicate the obturator nerve exiting the canal. Short white arrows indicate the implant. Black arrows point to the pubis symphysis. The distance between the implant and the canal is also shown in panels (**D** and **E**).

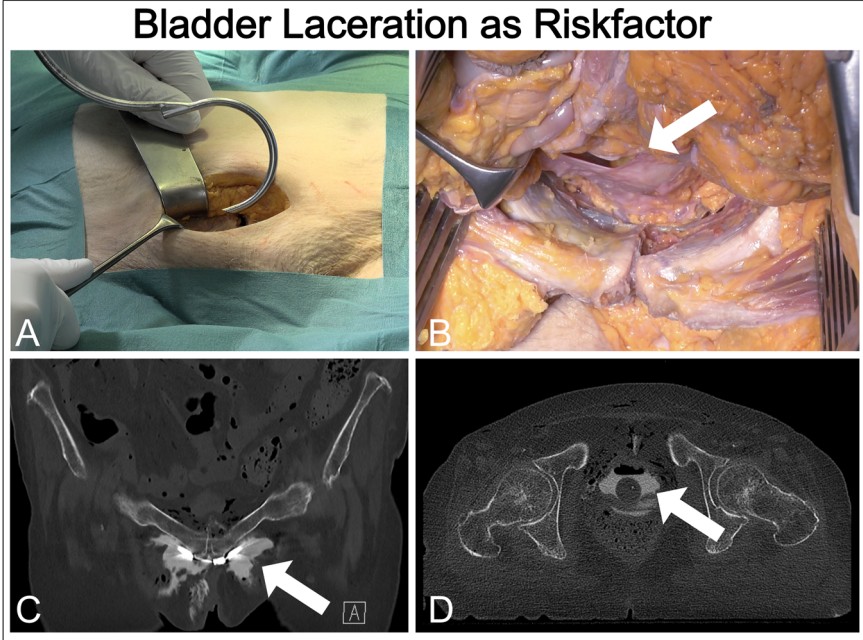

**Fig. 9 Risk of bladder laceration during surgical procedure.** Bladder laceration might result from vigorous application of an oversized curved cable passer during trans-obturator surgery with limited visibility. **A** Bladder laceration was observed in our cadaver study, where an oversized cable passer was used to transfer the cable from one side to the other. The bladder injury was confirmed by (**B**) visual inspection during preparation, and observation of (**C**) extravasation of contrast fluid around the pubis symphysis and (**D**) entrapped air in the bladder in computed tomography (CT) images obtained before full anatomical dissection.

with high tensile strength and confirmed a good ex vivo stability[38]. Böhler et al. applied suture buttons as alternative fixation in a comprehensive cadaver study and found no inferiority compared to plating[39]. Kiskaddon et al. confirmed feasibility of suture button fixation of the pubic symphysis in their cadaver study[40]. Cavalcanti-Kussmaul et al. explored suture tape fixation of the pubic symphysis and yielded promising results in a synthetic bone model test set-up and cadaver test model[13,41]. In contrast to the mentioned studies we decided to use a braided steel cable as tensioning device. Even so the use of modern suture material seems also reasonable. Further, the question arises whether the cable used here could be applied without the presented clamps. In a previous study we confirmed that trans-obturator cable fixation is feasible in a synthetic bone model. Preliminary clinical cases in which the technique was applied in combination with sympyseal plating shows promising results. However, we fear loosening of the cable when used without an anchoring device because the ongoing micromotion may cause the braided cable to cut into the bone[20].

In addition to the biomechanical analyses, we demonstrated safe placement of the implants next to the pubic symphysis using a trans-obturator approach. Our anatomical study identified a safe anatomical area for implant placement and highlighted structures at risk. Immediate hazards such as bladder laceration were described in detail and precautions to avoid such collateral injury were noted. Development of orthopedic implants such as those presented here is closely linked to advancements in implant manufacturing. Implant designs and manufacturing procedures become obsolete whenever better surgical techniques or fabrication processes emerge. The current standard treatment is SP-fixation; however, implant failure is often reported as previously mentioned[4,7,11,42–44]. Steel plates were originally intended to stabilize bony fractures until osseous healing could be completed; therefore, their suitability for stabilizing a dynamic, joint-like union such as the pubic symphysis is debatable. Screw detachment or loosening and plate breakage can occur when symphyseal

micromotion is high, which in our experience arises especially among non-compliant, elderly, or obese patients[11]. Our CCAO or CCAP designs may therefore be advantageous because they do not rely on screw fixation. Fabrication of these implants was enabled using a modern 3D printing technology, which has greatly enhanced metal-based orthopedic prototype development. The ease of 3D printing and its ability to realize complex design structures makes it ideal for implant research, especially using CMF. Ultra-fast production of a dedicated cable-clamp implant using patient-specific medical imaging data may be implemented in the future[45]. Although our implant geometries were designed based on synthetic bone models, they fitted well to the cadaver specimens so that scaling of the existing clamp sizes may also be sufficient. The synthetic binder poses no concerns for biocompatibility and stability because debinding and sintering removes all remnants; however, implant manufacture using more established techniques, such selective laser melting or electron-beam melting, is also an option because some are already FDA-approved[46].

The trans-obturator surgical approach is necessary to implant the CCAO and CCAP devices but not generally established in orthopedic surgery. Operating through the obturator foramen has been shown to be safe in other surgical specialties[47,48]. In our anatomical study, none of the cadavers had a ruptured symphysis and all had an intact rectus abdominis. Therefore, visualization through the symphysis into the true pelvis was impaired. Supportive cranial midline incision was needed to improve visual access but might be unnecessary under real conditions in which the pubic symphysis is gapping. As previously described by Hadeed et al., some important anatomical structures must be considered[49]. In particular, the spermatic cord and ligamentum suspensorium penis in men and the round ligament, clitoral body, and clitoral glans in women must be spared to prevent dyspareunia, erectile dysfunction, or infertility[50–53]. During our dissection, we identified no conflict with the abovementioned structures. The spermatic cord was sufficiently distant, but its

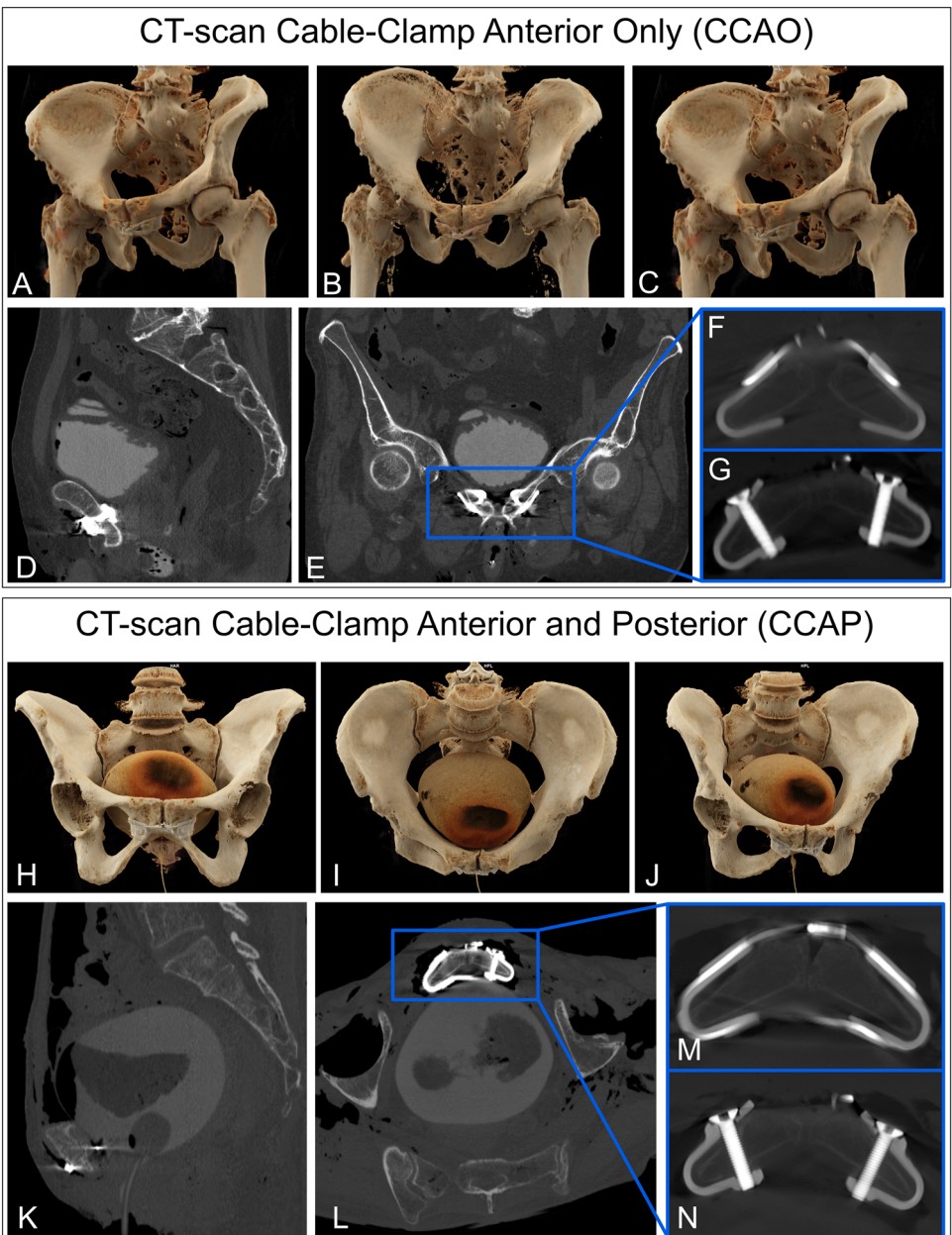

**Fig. 10 Implant evaluation by computed tomography.** Top: (**A–C**) CT images of the CCAO device implanted at the pubic symphysis. **D**, **E** Sagittal and axial CT images of the intact bladder filled with contrast fluid. **F**, **G** Enlarged images showing the good implant fit to the bone, anterior placement of the cable, and screws fixing the clamp to the bone. Bottom: (**H–J**) CT images of the CCAP and contrast-filled bladder. **K**, **L** Sagittal and axial CT images of the contrast-filled bladder containing a frozen blood clot. **M**, **N** Enlarged images showing the implant fit to the bone, cable surrounding the pubic symphysis, and screws fixing the clamp to the bone.

position in a trauma context might differ so reduction clamps or a cable passer should be used with caution[52]. Regarding the uriniferous structures, the urethra runs underneath the pubic symphysis. The Foley catheter could be palpated and was outside the operating area. Structures of the obturator canal are at the lateral border of the foramen and were usually several centimeters from the clamp. The trans-obturator approach may be considered an extension of previously described minimally invasive techniques such as the Fu-Liu approach[54]. To place the implant, a cable must be passed through muscles such as the pectineus, adductors, obturator externus and internus, and part of the levator ani. The risk of herniation is unknown.

The following limitations of our work should be considered: We used a single-leg stance model for our study because in our experience it mimics implant loosening of the pubic symphysis and clinical postoperative non-weightbearing. However, some authors reasonably argue that a bilateral-leg stance model better creates distraction forces at the pubic symphysis and shear forces through the sacroiliac joint[8,55–57]. Furthermore, the number of our cyclic loadings may have been inadequate. Six weeks of daily walking would amount to approximately 210,000 cycles, greater than the number tested here[58]. Our pre-tests were conducted using a larger number of cyclic loads during which diminishing differences were seen, but some disadvantages of the cable-clamp device might not have been apparent. Our synthetic pelvic model had no ligaments or muscular attachments, and the load vectors would differ under real-life conditions. Nevertheless, synthetic bone models are established and well-accepted precursors to

cadaver studies[36,59,60]. Screw and plate fixation in our study was performed without a drilling template that would otherwise improve conformity of the implantation[25,26]. Use of the CCAO and CCAP devices is limited to pubic symphyseal rupture, and fracture of the pubic rami disqualifies any type of cable-clamp fixation. Finally, we acknowledge that analogous approaches such as trans-osseous wiring of the symphysis have previously been described[18,19,61,62]. Successful biomechanical fixation has been achieved using various alternative techniques[13-16,20,41,55,63-66], but most have not entered clinical practice. We propose that the CCAO or CCAP devices may be of clinical value.

## Conclusions

Our biomechanical proof-of-concept study demonstrated that SP and cable-clamp fixation provide equivalent stabilities. The trans-obturator approach required for cable-clamp fixation represents a novel surgical procedure that conforms with the current trend toward less-invasive pelvic surgery. Clinical studies are needed to determine if CCAO or CCAP devices will produce better functional outcomes.

## Data availability

The authors declare that the source data supporting the findings of this study is available in Supplementary Data 1.

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

## Acknowledgements

The project was funded by the German Research Foundation (DFG JO1467/2-1) and Universitätsbund Würzburg (AZ 20-10). We thank Daniela Keller for support with statistical analysis and Anna Kellersmann for video documentation. We thank Edanz for editing a draft of this manuscript. This publication was supported by the Open Access Publication Fund of the University of Wuerzburg.

## Author contributions

M.C.J., C.F. and R.H.M. created the prototype. M.C.J. wrote the paper. M.C.J. and D.B. drafted the figures. M.C.J. and D.B. performed statistical analysis. D.B., M.C.J., P.H., F.G., S.H.D., and R.G.J. conducted the biomechanical tests. M.C.J., D.B., S.E. and C.K. performed anatomical preparation and explored the surgical approach. K.P. helped digital planning. J.P.G. and H.H. performed CT-diagnostic. M.C.J. is responsible for the conception and design of the manuscript. All authors approved the final version of the manuscript.

## Funding

## Competing interests

The Julius-Maximilians-University Würzburg (M.C.J., C.F., and R.H.M.) claimed a patent for the cable-clamp fixation device (International Publication Number: WO 2022/207348 A1). R.H.M. is consultant for Medartis Holding AG. All other authors have no competing interests.
