## [Peer Review File · Communications Medicine]

Reviewers' comments:

Reviewer #1 (Remarks to the Author):

The study focuses on new implants for pubic symphysis fixation, in comparison to symphyseal plating. The manuscript describes the 3D printing process, implantation techniques, and biomechanical evaluation. With these three components, the manuscript seems to be designed to support the device rather than provide a purely objective evaluation. For a research study, I would suggest focusing primarily on the biomechanical evaluation and improving the analysis.

1. Introduction: The first paragraph of the Introduction is too long and lacks a clear focus. The details on the 3D printing process seem excessive and are not all relevant to biomechanical evaluation. Existing literature on biomechanical evaluation of symphyseal plating is lacking. The results should not be summarize in the Introduction. I would recommend a first paragraph focused on the injury and limitations of symphyseal plating, a second paragraph focused on the existing literature on biomechanical evaluation of symphyseal plating, and third paragraph focused on the biomechanical hypothesis.
2. Hypothesis: The hypothesis is that cable-clamp fixation would be a superior alternative to symphyseal plating, but this is not tested. The analysis focuses on non-inferiority.
3. Biomechanical testing methods: Use of the synthetic pelvis models for this application needs more justification with additional description of the models and references to previous studies focused on pelvic injuries. References for alignment, force magnitudes and cycles used to represent single leg stance are also needed.
4. Biomechanical output: It is not clear to me how measurements related to vertical displacement of the load applicator are relevant to evaluation of the pubic symphysis. The measures at the pubic symphysis are more relevant. The accuracy of the 3D motion tracking system should be provided. The coordinate system used to measure separation should be described in detail.
5. Cadaveric models: I don't understand how the cadaver models could be tested without a support through the ilium.
6. Trans obturator approach: There does not seem to be any objectively measured output data for this part of the study.
7. Statistical analysis: What was the effect size used for the power analysis? What tests were used to compare data between the three groups for the synthetic models. Description of an ANOVA with post-hoc testing seems to be necessary.
8. Implant manufacturing and handling: This does not seem like necessary information.
9. Biomechanical analysis: No p-values are provided for any of the testing results.
10. Establishment of trans obturator approach: There is no data in this section of the Results.
11. Discussion: The Discussion lacks comparisons of the biomechanical data to the existing literature and lacks rigorous evaluation of how to interpret the results with respect to treatment of a ruptured pubic symphysis.

Reviewer #2 (Remarks to the Author):

Dear authors,

the article addresses a very important and repeatedly discussed issue about instability of the pubic symphysis. Cerclage systems have been used for this purpose, however with available anatomical plates the importance decreased. With the documented cases in your article you show the current problems with the existing implants, loosening, breakage and instability. The novel 3D printed implant in combination with a already available cable cerclage seems very promising for this purpose.

From the two different designs the CCAP look challenging when used in reality, although the data shows reduced values for displacement and higher values for stiffness. Regarding your data I guess you would suggest the CCAP design for a clinical purpose?

In your manuscript you describe the biomechanical testing up to 600 N, however in your data you only show 300N and 400N, please edit this in your m&m section. Did you also use a two leg stance set up. From my experience the pubic symphysis closes during single leg stance and opens when using a two leg stance, hence testing with both setups increases the information about the acting forces in the pubic symphysis.

One important question for the whole manuscript, why did you use these new implants at all? Why didn't you use the cable cerclage without your new implant? How will the cerclage act without the implant alone? Is the implant really necessary or just nice to have? Please add this to your discussion, if there is data available, please add this.

Reviewer #3 (Remarks to the Author):

I recommend publishing this excellent and interesting article after minor revisions as suggested above.

Thanks for the great manuscript.

The idea is novel and unique to my knowledge.

The research question is adequately addressed.

Statistical analysis is appropriate.

Please see the attached word doc. for the suggested minor revisions.

Review Jordan et al – Symphysis

Summary:

The underlying manuscript describes a biomechanical study on a new 3-d printed implant using a cable-wiring technique for traumatic separations of the pubic symphysis. It describes two different techniques and compares them to standard symphyseal plating. Both techniques show acceptable results comparable to ORIF.

Overall:

The article is very well written and covers an interesting topic in trauma surgery. Alternative techniques for symphyseal rupture are discussed a lot recently. The manuscript is especially interesting as it not only covers biomechanical testing but also shows how a new implant can be manufactured to address a specific anatomical part. Individual implant design using 3D printers as in this case could be the future of implant development according to biomechanical and anatomical needs.

Comments:

1. Line 60 ... at risk OF dislocation....

2. Line 67 Implant MANUFACTURING...

3. Line 137 What literature is the set up based on? Was it used in a similar way before? Here a similar single leg stance was used: Cavalcanti Kußmaul A, Greiner A, Kammerlander C, Zeckey C, Woiczinski M, Thorwächter C, Gennen C, Kleber C, Böcker W, Becker CA. Biomechanical comparison of minimally invasive treatment options for Type C unstable fractures of the pelvic ring. *Orthop Traumatol Surg Res.* 2020 Feb;106(1):127-133. doi: 10.1016/j.otsr.2019.09.032. Epub 2019 Dec 18. PMID: 31864961.

I suggest adding this or other references to refer to an established single leg stance model.

4. Line 306: Synthetic models are fine for biomechanical testing when the aim is to compare implants. As mentioned they are very well comparable and differences in the implants stability can be found. However human cadaver testing is always superior as the implant is supposed to be used in patients at one point. Here I suggest doing a second study with a greater case number after sample size calculation.

5. Line 351: The obturator approach seems quite invasive to me, I'm not sure if it is actually less invasive compared to the Pfannenstiel approach. There are less invasive ways to stabilize the pubic symphysis as you discussed, however I think we need new methods which should be further evaluated

Overall:

I recommend publishing this excellent and interesting article after minor revisions as suggested above.

Thanks for the great manuscript.

Point-to-point response

Title:

Solving Implant Failure in Symphyseal Plating: Development and Preclinical Evaluation of a Cable-Clamp Fixation Device for the Disrupted Pubic Symphysis

Manuscript ID: COMMSMED-22-0252-T

Reviewer # 1	Response	Edit in manuscript
The study focuses on new implants for pubic symphysis fixation, in comparison to symphyseal plating. The manuscript describes the 3D printing process, implantation techniques, and biomechanical evaluation. With these three components, the manuscript seems to be designed to support the device rather than provide a purely objective evaluation. For a research study, I would suggest focusing primarily on the biomechanical evaluation and improving the analysis.	Dear reviewer, we highly appreciate your valuable insight and attention to detail. We agree that the biomechanical analysis represents the most important part of our research, upon which the main focus should lie. However, we intend the manuscript to be a proof-of-concept study not limited to the biomechanical outcome. The evaluation of surgical feasibility is mandatory in justifying any further research, as well as of great interest to clinicians. While we understand the reasoning behind your argument, we would prefer to include some aspects of the surgical approach to appeal to clinician readers. We agree that the 3D-print section is overloaded. We have deleted redundant sections in the introduction, m&m, and discussion, and have changed the title. 3D-printing is of increasing interest in orthopaedic surgical research and its applying. This manuscript represents one of the pioneering examples of a metal-based 3D-print	Revision is marked throughout the text. We revised the title to maintain the focus on biomechanical aspects and not on 3D-printing form: "Solving Implant Failure in Symphyseal Plating: Development and Preclinical Evaluation of a Novel 3D-printed Fixation Device for the Disrupted Pubic Symphysis" to: "Solving Implant Failure in Symphyseal Plating: Development and Preclinical Evaluation of a Cable-Clamp Fixation Device for the Disrupted Pubic Symphysis"

	process for prototype development. The additive printing process is essential in the implants design, and currently the subject of controversial discussion in orthopaedic circles. With this in mind, we respectfully request the continued inclusion of this aspect in the final manuscript. As you pointed out, the manuscript aims to present an objective study instead of an advertisement for the device. We have revised our wording to reflect this. We hope you understand our standpoint and would be grateful for a further round of revision.	
1. Introduction: The first paragraph of the Introduction is too long and lacks a clear focus. The details on the 3D printing process seem excessive and are not all relevant to biomechanical evaluation. Existing literature on biomechanical evaluation of symphyseal plating is lacking. The results should not be summarize in the Introduction. I would recommend a first paragraph focused on the injury and limitations of symphyseal plating, a second paragraph focused on the existing literature on biomechanical evaluation of symphyseal plating, and third paragraph focused on the biomechanical hypothesis.	According to your recommendation we shortened our introduction and split it into three sections:  1) Symphyseal disruption and plate fixation 2) Biomechanical considerations of symphyseal plating 3) Hypothesis of an alternative fixation technique 	Please see our rewritten introduction. Page 2, line 54-100.
2. Hypothesis: The hypothesis is that cable-clamp fixation would be a superior alternative to symphyseal plating, but this is not tested. The analysis focuses on non-inferiority.	The hypothesis was incorrect. We have rewritten it to reflect non-inferiority testing.	Page 2, line 94: We hypothesized that cable-clamp fixation would be an equivalent alternative to SP fixation.

3. Biomechanical testing methods: Use of the synthetic pelvis models for this application needs more justification with additional description of the models and references to previous studies focused on pelvic injuries. References for alignment, force magnitudes and cycles used to represent single leg stance are also needed.	As you pointed out, the use of synthetic pelvis bone models is controversial and requires justification. We have included more references to reflect the wider use of the same synthetic bone models in other biomechanical laboratories. We added several new citations to justify the use of a single leg stance test set-up and the references for our test protocol.	We justify the use of synthetic bone models through its widely accepted use in biomechanical laboratories and its recent utilization in similar pelvic studies. This ensures the comparability of our results^{19,30-33}. Page 4, line 297: We used a single-leg-stance model in the biomechanical test setup as it is well-established, simple, reproducible, and is at minimal risk for confounders^{36,37}. This allows the application of vertical shear stress to the pubic symphysis without traction. Different biomechanical test set-ups were explored in pre-testing, such as the two-leg stance with simultaneous load or with alternating load, but all have separate limitations^{17,38}.
4. Biomechanical output: It is not clear to me how measurements related to vertical displacement of the load applicator are relevant to evaluation of the pubic symphysis. The measures at the pubic symphysis are more relevant. The accuracy of the 3D motion tracking system should be provided. The coordinate system used to measure separation should be described in detail.	We agree that the movement of the pubic symphysis is a more valuable parameter than vertical displacement of the whole construct. However, we argue that the vertical displacement indirectly provides important data about deformation of the anterior and posterior pelvic ring. We added a detailed explanation about the calibration of our 3D motion tracking system.	Page 4, line 314: Visual markers served as identifiers for the 12 megapixel camera and a reference block (140x22x32 mm) with three point markers was defined as coordinate system for the evaluation. This enabled the displacement of the individual visual markers to be measured in the X, Y, and Z axis. For measurement, the 30 visual measuring points were divided into the following point components: coordinate system, femur, right symphysis, left symphysis, right sacroiliac joint, and left sacroiliac joint. Among the several visual measuring points, we determined... Page 12, line 621:

	We also included a figure explaining the measurement of 3D-tracking.	Figure 6. The 3D-motion capturing. (A) The system was calibrated and movement measured in the x-, y-, and z-axis. (B) Horizontal movement of the x-axis was less important in our single leg stance test because there was no tension at the pubic symphysis. (C) Vertical movement along the y-axis was the most reliable parameter for our testing because it documented the vertical shear stress. (D) Movement in the anterior-posterior direction along the z-axis was documented but less relevant.
5. Cadaveric models: I don't understand how the cadaver models could be tested without a support through the ilium.	Good point. In our pre-tests, both the cadaver specimens and the synthetic bone models we used did not include the ileum attachment, and the cadaver specimens were better able to withstand loads of 400N than the synthetic bone models. We hypothesized that this is due to the preserved posterior sacroiliac ligament, absent in the synthetic bone model. When testing cadaver bone with the ileum attachment, we found that movement at the anterior pelvic ring was very limited. To ensure a reliable outcome of the movement at the pubic symphysis we therefore decided upon the described test set-up. We see each test protocol as individual. Other biomechanical studies confirm our theory that an ileum attachment is not mandatory (Agarwal Y. et al. or Stuby FM. et al.). Agarwal Y. et al. Two-leg alternate loading model - A different approach to biomechanical investigation of fixation methods of the	Page 5, line 336: In this special setting we decided against the ileum attachment in order to freely observe movement at the anterior pelvic ring.

	injured pelbic ring with focus on the pubic symphysis. Journal of Biomechanics 2014 Stuby FM. et al. Influence of flexible fixation for open book injury after pelvic trauma - A biomechanical analysis. Clinical biomechanics 2013	
6. Trans obturator approach: There does not seem to be any objectively measured output data for this part of the study.	We agree that there is no objective parameter. This section was intended as a description of a surgical approach. The variability of the cadavers makes obtaining objective parameters difficult. Despite this, we feel that including the surgical approach is crucial to our manuscript. We have removed redundant information according to your recommendation.	Several sections deleted throughout the text.
7. Statistical analysis: What was the effect size used for the power analysis? What tests were used to compare data between the three groups for the synthetic models. Description of an ANOVA with post-hoc testing seems to be necessary.	Power Analysis: Analysis was performed a priori using data from our previous studies (G*Power software). Power analysis was done for different parameters such as stiffness or total displacement. 0.1-0.3 small effect 0.3-0.5 moderate effect > 0.5 large effect ANOVA: We compared the means and added this data in our revised manuscript. We did not include this data initially, as our statistician recommended the use of only the Equivalence-and-Noninferiority test. No significant difference was measured in the ANOVA, but the information does not securely confirm equivalence.	Page 5, line 358: A power analysis was performed using a power of 80% and a significance level of 5%, which showed that the sample size was adequate. The results are presented as mean values with standard deviations. All data underwent statistical analysis for normal distribution using the Shapiro–Wilk test. Analysis of variance was used to compare the means. A post hoc test was unnecessary because no difference was measurable. A P value of less than 0.05 was considered statistically significant. Page 15, line 711: Table 4. Statistical analysis of the synthetic bone models tested by ANOVA. The results are presented as mean values with standard

		deviation. All data underwent statistical analysis for normal distribution using the Shapiro–Wilk test (*no normal distribution). Analysis of variance was used to compare the means. A post hoc test was not necessary because no difference was measurable. A P value of less than 0.05 was considered statistically significant. Absence of a significant difference cannot be interpreted as equivalence of the implants. Therefore, the non-inferiority testing was necessary.
8. Implant manufacturing and handling: This does not seem like necessary information.	We deleted redundant information. However, we believe basic information of 3D-printing may be of interest to readers who wish to use the presented technique.	Several sections deleted.
9. Biomechanical analysis: No p-values are provided for any of the testing results.	The Two-one-sided t-tests (TOST) do not generate p-values. The results are presented as equivalence ranges. We have included more detailed information at the end of our point-to-point response (please see text at the end)	
10. Establishment of trans obturator approach: There is no data in this section of the Results.	As we mentioned before we view this as additional technical information.	
11. Discussion: The Discussion lacks comparisons of the biomechanical data to the existing literature and lacks rigorous evaluation of how to interpret the results with respect to treatment of a ruptured pubic symphysis.	Your concern is justified. We revised our discussion section and added biomechanical data as well as information on how to interpret the results. Once again, we are grateful for your work as reviewer. Kind regards, the authors.	Page 8, line 488: Comparison of our results to the current literature confirms a current pursuit of alternative fixation devices. Most recently, Hinz et al. developed a flexible implant by combining plates with a fiber cord with high tensile strength, and confirmed a high ex vivo stability⁴³. Böhler et al applied suture buttons as alternative fixation in a comprehensive

		cadaver study and found no inferiority compared to plating⁴⁴. Kiskaddon et al. confirmed feasibility of suture button fixation of the pubic symphysis in their cadaver study⁴⁶. Cavalcanti-Kussmaul et al. explored suture tape fixation of the pubic symphysis and yielded promising results in a synthetic bone model test set-up and cadaver test model^{19,45}. In contrast to the mentioned studies, we decided to use a braided steel cable as tensioning device. Despite this, the use of modern suture material seems also reasonable.
Reviewer #2	Response	Edit in the Manuscript
The article addresses a very important and repeatedly discussed issue about instability of the pubic symphysis. Cerclage systems have been used for this purpose, however with available anatomical plates the importance decreased. With the documented cases in your article you show the current problems with the existing implants, loosening, breakage and instability. The novel 3D printed implant in combination with a already available cable cerclage seems very promising for this purpose.	Dear reviewer, we appreciate your comments and are grateful for your time. Indeed, cerclage fixation is not novel, and has long been replaced by the use of plates. We experienced complications with these plates especially in the elderly population, and may revive the cerclage through the new technique. Through the revision process we have made several changes to our manuscript. Despite this, the main findings remain.	Several changes throughout the text.
From the two different designs the CCAP look challenging when used in reality, although the data shows reduced values for displacement and higher values for stiffness. Regarding your data I guess you would suggest the CCAP design for a clinical purpose?	Indeed, CCAP will be more challenging to implant. Despite this, we tend toward the CCAP for clinical application.	No changes made.
In your manuscript you describe the biomechanical testing up to 600 N, however	600 N refers to our pretests. The final testing was done with 300 and 400 N. We	Page 4, line 306: A series of pre-tests were performed, using up to 12

in your data you only show 300N and 400N, please edit this in your m&m section. Did you also use a two leg stance set up.	highlighted this information in our M&M section. In the current project we used single leg stance test as it is simple and reliable. The two leg stance with alternating load is likely a more realistic simulation, but data reflecting this cannot be provided at the moment.	000 test cycles and load levels up to 600 N, until a final test protocol was defined.
From my experience the pubic symphysis closes during single leg stance and opens when using a two leg stance, hence testing with both setups increases the information about the acting forces in the pubic symphysis.	A single-leg stance puts vertical shear stress on the pubic symphysis. A two-leg stance results in traction instead. In our pretests we were able to induce loosening of symphyseal plates through vertical load application in a one leg stance test. We believe that instead of traction, vertical shear stress is responsible for loosening.	No changes made.
One important question for the whole manuscript, why did you use these new implants at all? Why didn't you use the cable cerclage without your new implant?	We tested the cable cerclage without the new implants in a previous study where we demonstrated its feasibility. Several of our clinical cases back these findings as well. However, we have also observed cases in which the cerclage cut into osteoporotic bone. In our opinion, the anchoring element could remove this risk of loosening and improve long-term stability. (Jordan et al. Trans-obturator cable fixation of open book pelvic injuries. Sci Rep 2021).	Page 8, line 488: Comparison of our results to the current literature confirms a current pursuit of alternative fixation devices. Most recently, Hinz et al. developed a flexible implant by combining plates with a fiber cord with high tensile strength, and confirmed a high ex vivo stability⁴³. Böhler et al applied suture buttons as alternative fixation in a comprehensive cadaver study and found no inferiority compared to plating⁴⁴. Kiskaddon et al. confirmed feasibility of suture button fixation of the pubic symphysis in their cadaver study⁴⁶. Cavalcanti-Kussmaul et al. explored suture tape fixation of the pubic symphysis and yielded promising results in a synthetic bone model test set-up and cadaver test model^{19,45}. In contrast to the mentioned studies, we decided to use a braided

		steel cable as tensioning device. Despite this, the use of modern suture material seems also reasonable. The question arises whether the cable used here could be applied without the presented clamps. In a previous study we confirmed that trans-obturator cable fixation is feasible in a synthetic bone model. Preliminary clinical cases in which the technique was applied in combination with symphyseal plating show promising results. However, we fear loosening of the cable when used without an anchoring device, because the ongoing micromotion may cause the braided cable to cut into the bone ²⁶.
How will the cerclage act without the implant alone? Is the implant really necessary or just nice to have? Please add this to your discussion, if there is data available, please add this.	See comment above and our revised discussion.	See changes above.
Reviewer #3	Response	Edit in Manuscript
I recommend publishing this excellent and interesting article after minor revisions as suggested above. Thanks for the great manuscript.	Thank you for your insight. We addressed all your concerns and comments as well as those of the other reviewers. (The comment on the left is all we obtained from you. No Word-document was available)	Several changes throughout the text.

Additional information about our statistic of the non-inferiority testing:

Data of the of synthetic bone model study underwent an Equivalence-and-Noninferiority Testing. The confidence intervals represent the preferred statistical method for testing for non-inferiority (= equivalence or superiority). In order to maintain a significance level of 5%, both the lower (CU) and the upper confidence limit (CO) must have a one-sided confidence level (probability of confidence) of 95% own. The equivalence limits are given as the symbols $-\epsilon_1$ (lower equivalence limit) and ϵ_2 (upper equivalence limit) and define the equivalence range Δ , also called the irrelevance range. Here, ϵ_1 and ϵ_2 are positive numbers that determine the range of deviation from the reference value that is still acceptable. For the present statistical evaluation, a relative equivalence limit of 15% deviation from the symphysis plate (reference group of the entire

evaluation) was chosen and preferred to an absolute value based on two different force levels (300 N, 400 N). Accordingly, $\varepsilon = 0,15 * M_1$ applies.

The 95% confidence interval is given by:

$$M_1 - M_2 \pm t_{n_1+n_2-2;1-\alpha} * S_D * \sqrt{\frac{1}{n_1} + \frac{1}{n_2}}$$

with the one-tailed quantile of the t-distribution $t_{n_1+n_2-2;1-\alpha} = 1,734$ as well as the standard deviation

$$S_D = \sqrt{\frac{(n_1-1) * s_1^2 + (n_2-1) * s_2^2}{n_1 + n_2 - 2}}$$

M_1 is the mean value of the group to be compared (cable-clamp anterior only or cable-clamp anterior and posterior), whereas M_2 represents the mean value of the symphysis plate (reference group). n_1 or n_2 are the corresponding sample sizes ($n_1 = n_2 = 10$) and s_1 is the standard deviation of the group to be compared, s_2 that of the symphysis plate.

This results in the lower confidence limit

$$C_U = M_1 - M_2 - 1,734 * \sqrt{\frac{(n_1 - 1) * s_1^2 + (n_2 - 1) * s_2^2}{n_1 + n_2 - 2}} * \sqrt{\frac{1}{n_1} + \frac{1}{n_2}}$$

and the upper confidence limit

$$C_O = M_1 - M_2 + 1,734 * \sqrt{\frac{(n_1 - 1) * s_1^2 + (n_2 - 1) * s_2^2}{n_1 + n_2 - 2}} * \sqrt{\frac{1}{n_1} + \frac{1}{n_2}}$$

The assumption of non-inferiority requires a determination as to whether higher or lower measured values are to be regarded as more stable: If higher values are considered to be more stable, as is the case for the stiffness parameter, then non-inferiority can be assumed if $C_U > -\varepsilon 1$. On the other hand, if lower values are considered to be more stable, the assumption of non-inferiority requires that $C_O < \varepsilon 2$. This applies to the parameters of the material testing machine for peak-to-peak displacement, total displacement, plastic deformation and $D_{\min 5000} - D_{\max SZ10}$. Smaller values are also considered more stable for the $|R1-L1|$ -displacement in the Y-direction in the optical measuring system. Thus, non-inferiority is also given here if $C_O < \varepsilon 2$.

The determination of confidence intervals requires a normal distribution of the measured values, which was checked using the Shapiro-Wilk test. In addition to the normal distribution of the data, the determination of the confidence intervals requires homogeneity of variance, for which the Levene test was used. Normal distribution and homogeneity of variance of the measured values of the two groups to be compared is a necessary condition for a test for non-inferiority. If this condition was not primarily met, the data of both test groups were transformed using the square root in order to obtain normally distributed, homogeneous variance values. Boundaries transformed in this way were then retransformed to subsequently test for non-inferiority.

REVIEWERS' COMMENTS:

Reviewer #1 (Remarks to the Author):

The authors have made substantial changes to address my previous comments. This review addresses the revisions made by the authors.

Overall: My personal preference would be a study focused specifically on the biomechanics of symphyseal plating, rather than a proof of concept for several aspects of the cable clamp fixation device, but I can accept the broader approach if it is valuable to clinicians.

Introduction: The format of the Introduction is improved. Can the authors quantify the rate of implant failure and gapping?

Hypothesis: The hypothesis has been rewritten to focus on non-inferiority. This new hypothesis brings up the concern that a device system that only shows non-inferiority compared to the symphyseal plating will not reduce the risk of gapping. Please clarify how a non-inferior system could improve outcomes with respect to the failure modes described in the Introduction.

Biomechanical output: The authors argue in the response that the vertical displacement data provides important information about deformation of the anterior and posterior pelvic ring in the response. The relevance of this data is not described in the manuscript, however. The accuracy of the motion tracking still has not been provided. It should be much smaller than the displacements on the order of 1 mm.

Reviewer #2 (Remarks to the Author):

Dear authors,
thank you for revising the manuscript.
I have no more requests for revision.

Reviewer #3 (Remarks to the Author):

All reviewer comments were addressed and answered.
I suggest accepting the manuscript.

Point-to-point response

Title:

Development and Preclinical Evaluation of a Cable-Clamp Fixation Device for a Disrupted Pubic Symphysis

Manuscript ID: COMMSMED-22-0252-T

Reviewer # 1	Response	Edit in manuscript
Introduction: The format of the Introduction is improved. Can the authors quantify the rate of implant failure and gapping?	Dear reviewer, we highly appreciate your valuable second review. We included the information about implant failure.	Page 3, line 93: Some patients present only with mild radiologically verifiable plate loosening with an acceptable functional outcome (30-75%), whereas others require surgical revision and have a poorer outcome (3%-11%)⁵⁻¹¹.
Hypothesis: The hypothesis has been rewritten to focus on non-inferiority. This new hypothesis brings up the concern that a device system that only shows non-inferiority compared to the symphyseal plating will not reduce the risk of gapping. Please clarify how a non-inferior system could improve outcomes with respect to the failure modes described in the Introduction.	Good point. Regarding implant failure, non-inferiority alone does mean superiority. The aim of our study was to analyze implant devices with similar effectiveness to those currently used (SP-fixation), but with improved risk profiles such as fewer implant loosening, better surgical application, and lower risk of gapping. We agree that so far, our biomechanical study does not confirm superiority. We mention this aspect in the discussion section.	Page 4, line 130: We hypothesized that cable-clamp fixation would be a possible alternative to SP fixation. We conducted an equivalence and non-inferiority testing to proof effectiveness of new the implant devices compared to SP-fixation but with better properties such as fewer risk of implant failure or less invasive implantation.
Biomechanical output: The authors argue in the response that the vertical displacement data provides important information about deformation of the anterior and posterior pelvic ring in the response. The relevance of this data is not described in the manuscript, however. The accuracy of the motion tracking still has not been	It is an important point. The role of vertical displacement is now mentioned under M&M. From the numerous visual measuring points, we determined those on either side of the symphysis to be most relevant. movement in this area is referred to as symphyseal displacement (mm).	Page 5, line 210: Calibration was performed before each test and accuracy was confirmed (0.3-0.03 mm). Among the several visual measuring points, we determined those on either side of the symphysis to be most relevant, and we referred to the separation as

provided. It should be much smaller than the displacements on the order of 1 mm.	Further we added: " This parameter allows indirect measurement of movement at the pelvic ring." We added information about the accuracy as provided by the company, ARAMIS System accuracy 0,03 mm. Kind regards The authors	symphyseal displacement (mm). This parameter allows indirect measurement of movement at the pelvic ring.
---	---	---

Reviewers' comments:

Reviewer 1 : The authors have addressed my concerns adequately.